# A Review of Applications of Solid-State Nuclear Magnetic Resonance (ssNMR) for the Analysis of Cyclodextrin-Including Systems

**DOI:** 10.3390/ijms24043648

**Published:** 2023-02-11

**Authors:** Anna Helena Mazurek, Łukasz Szeleszczuk

**Affiliations:** 1Department of Organic and Physical Chemistry, Faculty of Pharmacy, Medical University of Warsaw, Banacha 1 Str., 02-093 Warsaw, Poland; 2Doctoral School, Medical University of Warsaw, Żwirki i Wigury 81 Str., 02-093 Warsaw, Poland

**Keywords:** cyclodextrin, ssNMR, drug carrier

## Abstract

Cyclodextrins, cyclic oligosaccharides composed of five or more α-D-glucopyranoside units linked by α-1,4 glycosidic bonds, are widely used both in their native forms as well as the components of more sophisticated materials. Over the last 30 years, solid-state nuclear magnetic resonance (ssNMR) has been used to characterize cyclodextrins (CDs) and CD-including systems, such as host–guest complexes or even more sophisticated macromolecules. In this review, the examples of such studies have been gathered and discussed. Due to the variety of possible ssNMR experiments, the most common approaches have been presented to provide the overview of the strategies employed to characterize those useful materials.

## 1. Introduction

Cyclodextrins (CDs) belong to the group of cage molecules; that is, the core of their structure is composed of a hydrophobic cavity that can trap or encapsulate other substances. Those remarkable encapsulation properties lead to a “host–guest” type relationship that can modify or improve the physical, chemical, or biological characteristics of the guest molecule. While the most commonly used CDs are α, β, and γ that contain 6, 7, and 8 glucose subunits, respectively, the large-ring cyclodextrins (LR-CD), composed of nine to more than several hundred units, are also being studied and utilized [1]. Further, apart from native (non-substituted) CDs, their derivatives have found important uses in various fields, such as pharmacy, cosmetics, biomedicine, textiles, and food domain due to their unique properties [2]. In addition to two-component inclusion complexes consisting of a guest molecule and a CD, polymer additives and CDs covalently linked to polymers are also of current interest. The biodegradability, biocompatibility, and versatility of CDs and CD-based materials extend their applications to new areas every year. CDs are commonly used in pharmaceutical formulations, as they increase the solubility of poorly soluble drugs and protect substances against external factors, such as light, humidity, and heat [3]. CDs can mask unpleasant smells or flavors of drugs, which is especially important in formulations dedicated to children [4]. More than 100 original drugs are currently being manufactured with CDs as excipients [5,6,7].

Interactions between CDs (host) and guest molecules may yield a stable complex with a high equilibrium constant. It is not surprising, then, that the number of newly obtained cyclodextrin host–guest complexes is continually increasing. While host–guest CD complexes can be developed as solution-phase formulations, they also have potential benefits as solids. Processes based on co-grinding, kneading, melt-spinning, spray-drying, lyophilization, coprecipitation, and solvent evaporation for the manufacture of CDs complexes have been extensively reported in the literature and in patent publications. In many of those cases, the method of obtaining of the complex had a major influence on the structure and properties of the product, leading either to amorphous or crystalline phases, sometimes even differing in the host:guest molecular ratio.

However, only a small amount of CD complexes is being reported together with their corresponding crystal structures. This is caused by the fact that most of those complexes are either amorphous or polycrystalline, and even for the crystalline complexes with well-established stoichiometry, it is usually very hard to obtain a crystal of a size suitable for single-crystal X-ray measurements [8,9]. However, in order to fully understand the aforementioned changes resulting upon complexation, knowledge of the molecular structure of the CD-based materials is essential. To achieve this goal, that is, to understand the structure and dynamics of the CD-based materials, multiple analytical and computational methods have been applied. The latter ones, including the quantum mechanics calculations [10] and molecular dynamics simulations [11], have been recently reviewed by us. From the experimental ones, likely the most commonly ones used in the analysis of CD-based materials in solid state are Fourier-transform infrared spectroscopy (FT-IR) and powder X-ray diffraction (PXRD), together with the thermo-analytical techniques, such as differential scanning calorimetry (DSC) and thermogravimetric analysis (TGA). Those methods have been discussed with regard to the CD dimer complexes in the review article from 2015 [12]. However, the application of solid-state nuclear magnetic resonance (ssNMR) can provide information unobtainable by any other method. In particular, ssNMR can provide the information on orientation of the guest molecule inside the cavity and the complex stability in the solid state. It also enables the quantitative analysis of the phases, especially the complexed and non-complexed guest molecules. In addition, this technique allows for the study of the local molecular dynamics of a guest molecules and the nature of intermolecular interactions between the host and the guest.

Therefore, this review aims to gather, discuss, and summarize the results of the studies addressing the ssNMR analysis of cyclodextrin-including systems. Due to the variety of ssNMR experiments that can be applied in this field, the most commonly used approaches have been summarized and described in more detail. Our goal is to familiarize the researchers who are investigating the synthesis and analysis of CD complexes with the benefits that can result from the application of ssNMR. Moreover, the examples discussed in this review can help to properly set up the ssNMR experiments and properly analyze the obtained results.

## 2. An Overview of the Methods Used in the Analysis of CD Complexes

Most of the methods applied for characterization of CD complexes are performed in solution phase. Solution-state NMR is a popular tool for measuring dissociation constants, determining stoichiometry, and elucidating the structure of CD complexes. UV−vis spectroscopy is also a popular approach for measurement of dissociation constants and relies on spectral changes upon inclusion, while fluorescence spectroscopy can make use of both spectral changes and fluorescence quenching effects. Nonetheless, methods for studying the CD-based materials directly in the solid state are limited.

Differential scanning calorimetry (DSC) can be used to confirm the occurrence of inclusion, especially when comparing the curves registered for the physical mixture and complex sample. However, since the complexation yield is usually lower than 100%, leading to the presence of free guest and host in the sample, the analysis of the DSC of CD complexes is usually a challenging task [13]. In addition, DSC does not provide any information about the structure or dynamics of the complex.

Powder X-ray diffraction (PXRD) is a relatively easy to perform analysis that can be used to study the materials, regardless of their crystallinity; in contrast to single crystal X-ray diffraction, it does not require the crystal of a proper size for the measurement. However, while the PXRD can be used to detect the new solid phase and can even be used for various quantitative analyses, PXRD cannot readily detect amorphous nonincluded (“free”) guest that may be present in complex materials [14]. In addition, if the obtained CD-based product is amorphous, PXRD cannot provide much information on it. Assuming the most optimistic scenario—presence of only one solid phase—PXRD diffractogram can be used determine the unit cell dimensions and crystal space group if the complex is a crystalline one [15]. While for some small molecular crystals it is possible to determine the positions of atoms in the unit cell, i.e., through Rietveld refinement, so far, this method has not been applied for the CD-containing systems.

Additionally, FT-IR and Raman spectroscopies are two complementary methods that can be used to study the CD-containing systems [16]. Through careful analysis of the recorder spectra, it is possible to confirm the inclusion process and, in some cases, determine the functional groups of the guest that interact with the CD molecule. However, due to the overlapping of the signals and highly labile nature of the guest, resulting in the static or dynamic disorder, it is usually very difficult to undoubtedly confirm the obtained results [17].

## 3. ssNMR Approaches for the Analysis of Cyclodextrin Complexes

Solid-state NMR is a powerful tool used in the study of all types of materials and phenomena. Therefore, it is not surprising that it has been successfully applied in the analysis of CD-based materials and complexes as well (Figure 1).

As expected, the review of the published works (Table 1) clearly shows that the most common type of the ssNMR experiment applied in the studies of the CD-based materials is the ^13^C cross-polarization magic-angle spinning (CP MAS) analysis, which is an obvious choice for the molecular solids and their mixtures. The CP method is based on heteronuclear dipole interactions; therefore, it is sensitive to internuclear distances and molecules mobility and, hence, can be used to monitor the molecular dynamics of complex systems. This method allows for the achievement of high-resolution ^13^C NMR spectra of solid materials, which provide information on the chemical and structural features of the samples being studied. To detect changes in crystallinity and the formation of inclusion complexes, the most relevant spectral features to be considered include the evaluation of signal splitting and changes in chemical shifts and linewidths of the detected ^13^C NMR signals (Figure 2 and Figure 3). Physical mixtures are often made for comparison to definitively confirm the inclusion process as well as to provide the information on the type of interaction between the guest and the host molecules.

Apart from the ‘standard’ 13C CP MAS experiment in the reviewed literature, successful application of more sophisticated experiments can be found as well. Below, some of those approaches will be summarized, followed by the description of the most interesting cases.

**Table 1 ijms-24-03648-t001:** General information regarding ssNMR application for the examination of the CD-including systems found in the articles published in period 2012–2022. Abbreviations used in the table: SEM (scanning electrine microscopy), XPS (X-ray photoelectron spectroscopy), XRD (X-ray diffraction), PXRD (Powder X-ray diffraction), TGA (thermogravimetric analysis), HR-TEM (high-resolution transmission electron microscopy), HPLC (high-performance liquid chromatography), ATR-FT-IR (attenuated total reflection Fourier-transform infrared spectroscopy), BET (Brunauer–Emmett–Teller method), DTA (differential thermal analysis), DLS (dynamic light scattering), EDS (energy-dispersive X-ray spectroscopy), WAXD (wide-angle X-ray diffraction), SR-FT-IR (synchrotron radiation Fourier-transform infrared spectromicroscopy, EIS (electrochemical impedance spectroscopy), SAXS (small-angle X-ray scattering), ITC (isothermal titration calorimetry), MALDI-TOF-MS (matrix-assisted laser desorption/ionization time-of-flight mass spectrometry).

CD	Guest	ssNMR Details	Reason for ssNMR Application	Other Experiments Used Alongside	Publication Year	Reason for Making a Complex with CD	Reference
novel β-CD based on N-halamine antimicrobial copolymer	sodium hydrochlorite	^13^C MAS	structure characterization	SEM, XPS, XRD, TGA, and DSC	2014	antimicrobial CD acetate nano-fibres	[19]
β	2-hydroxy-1-naphthoic acid	^13^C, ^1^H	structure characterization	UV-Vis, FT-IR, NMR, XRD, SEM	2016	sensory device for Ag cation	[20]
polyamino-CD	Ag(I) cation	^13^C CP MAS	structure characterization	TGA, FT-IR, ssNMR, SEM, HR-TEM	2019	nanosponge, sensory device for Ag(I) cation	[21]
succinyl- β	albendazole	^13^C, ^15^N MAS	structure characterization	ROESY and 1H NMR in solution	2018	potential drug carrier enhancing drug’s low aqueous solubility, spray-drying technique effectiveness	[22]
β, methyl- β, hydroxypropyl-β, citrate- β	albendazole	^13^C, ^15^N MAS	structure characterization, loss of crystallinity after spray-drying, explanation for differences in drug solubility enhancement found with different CDs	-	2015	potential drug carrier enhancing drug’s low aqueous solubility, spray-drying technique effectiveness	[23]
β, hydroxypropyl- β	1,2,4-thiadiazole derivative	^13^C MAS	structure characterization, explanation of differences in stability of different CD complexes	DSC, TG, PXRD, FT-IR, hot-stage microscopy, solubility measurements	2017	potential drug carrier enhancing drug’s low aqueous solubility	[24]
β	ciprofloxacin, doxorubicin, paclitaxel	^13^C MAS	confirmation of CD grafting, complex characterization	SEM, PXRD, FT-IR, HPLC	2019	nanocarrier, CD grafted on bacterial cellulose nanowhiskers	[25]
β	cinnamon and oregano essential oils	^13^C MAS	structure characterization	TGA	2017	nanofiber for packaging systems	[26]
β, methyl-β, hydroxypropyl- β	benznidazole	^13^C CP MAS	structure characterization	-	2015	potential drug carrier enhancing drug’s low aqueous solubility	[27]
β	benzoguanamine	^13^C, ^1^H MAS	structure characterization	UV-Vis, fluorescence, FT-IT, mass spectrometry	2018	chomosensor for selective Ce^4+^ sensing	[28]
β	bisacodyl	^13^C MAS	structure characterization, comparison of the complexes obtained from 3 different preparation methods	FT-IR, PXRD, TGA-DSC, SEM, 1H and ROESY NMR in solution	2015	potential drug carrier	[29]
β	bisphenol	^13^C dipolar-decoupled MAS	structure characterization	FT-IR, DOSY NMR in solution	2013	CD-polymer, pollutant-removing agent	[30]
2-hydroksypropyl- β	candesartan, candesartan cilexetil	^13^C CP MAS	structure characterization	mass spectrometry, HPLC	2019	potential drug carrier enhancing drug’s low aqueous solubility	[31]
per(3,5-dimethyl)phenylcarbamoylated-b	benzene homologues and phenylamine analogs	^13^C MAS	attachment of the CD moiety on the silica	FT-IR	2018	enantioseparation for isoxazolines, flavonoids, and β-blockers, polymer	[32]
β	ciprofloxacin hydrochloride	^13^C CP MAS	confirmation of CD being grafted onto cellulose fibers	FT-IR	2014	cyclodextrin-grafted cellulose fibers for antimicrobial products	[33]
β	-	^13^C MAS	confirmation of CD being grafted onto cellulose fibers	polarized optical microscopy, FT-IR, TGA	2012	cyclodextrin-grafted cellulose fibers for antimicrobial products	[34]
β	bisphenol A	^13^C MAS	structure characterization	FT-IR	2013	carboxylmethylcellulose-based hydrogel for toxin removal	[35]
β	acetylsalicylic acid	^13^C MAS	structure characterization	FT-IR	2014	CD-fraften carboxyheksyl chitosan hydrogels, biodegradable active material with controlled drug release	[36]
β	ketoconazole	^13^C CP MAS	structure characterization, confirmation of CD being grafted onto cellulose fibers	ATR-FT-IR, TGA-DSC	2017	citric-acid-crosslinked b-cyclodextrin/hydroxyethylcellulose hydrogel films for controlled delivery of poorly soluble drugs	[37]
β	ketoconazole	^13^C CP MAS	structure characterization, confirmation of CD being grafted onto cellulose fibers	ATR-FT-IR, TGA-DSC	2017	citric-acid-crosslinked cyclodextrin/carboxymethylcellulosehydrogel films for controlled delivery of poorly soluble drugs	[38]
β	ketoconazole	^13^C MAS	structure characterization	ATR-FT-IR, TGA-DSC, SEM	2016	citric-acid-crosslinked cyclodextrin/hydroxypropylmethylcellulosehydrogel films for hydrophobic drug delivery	[39]
	salicylic acid, salicylamide, piroxicam, hydrocortisone	^13^C CP MAS	structural changes under influence of raising temperature	PXRD, 1H NMR in solution	2017	CD columns in poly-ethylene–polyrotaxannes	[40]
β	sodium perfluorooctane	^13^C CP, DP, ^1^H, ^19^F MAS	structure characterization	DSC, TGA, FT-IR, PXRD	2015	toxin-removing agent	[41]
β	curcumin	^13^C, ^29^Si MAS	structure characterization	XRD, FT-IR, XPS, BET, TG, DTA, TEM, DLS	2015	nanocarrier, potential drug carrier enhancing drug’s low aqueous solubility, pH responsive ‘‘gate’’ when functionalized on the surface of mesoporous silica	[42]
β	diazepam	^1^H MAS	structure characterization, number of water molecules in the cavity	-	2014	drug carrier, lipophilic ligand transporter water compartments in the cell	[43]
β	1,4-diazepine derivatives	^13^C CP MAS	structure characterization	TGA, IR	2020	nanocarrier, potential drug carrier enhancing drug’s low aqueous solubility	[44]
γ	doxorubicine	^1^H -^1^H, ^13^C -^27^Al 2D MAS	structure characterization	-	2021	metal–organic frameworks nanoparticles with engineered cyclodextrin coatings	[45]
β	essential oils	^13^C CP MAS	structure characterization, comparison of different complexation techniques	FT-IR, DSC-TGA, PXRD	2019	antimicrobial sachets used as preservatives for foods	[46]
β	estrogen (estradiol, bisphenol A) and metal (zirconium) pollutants	^13^C MAS	structure characterization, adsorption mechanism	FT-IR, EDS	2019	Zr(IV)-cross-linked carboxymethyl-CD bifunctional adsorbent, toxin-removing agent	[47]
γ	fisetin	^13^C CP MAS	structure characterization	PXRD, Raman, TGA	2017	potential drug carrier enhancing drug’s low aqueous solubility	[48]
β	furosemide polymorphs	^13^C MAS	structure characterization	PXRD, SEM	2016	drug carrier, alternative matrices that improve physicochemical properties	[49]
α, β	CO2, N2	^13^C, ^15^N MAS	ligand incorporation	FT-IR, XPS, SEM, gas sorption porosimetry, 13C NMR in solution	2019	polyurethane aerogels, dessicant	[50]
β	hydrofluoroether	^13^C CP, ^19^F MAS	structure characterization, changes with temperature change	TGA, TG-MS, WAXD	2012	removing agent	[51]
α	--	^13^C MAS	structure characterization	ATR-FT-IR, transmission electron microscopy	2018	alkyl chains grafted on polysaccharides and CDs forming platelets used as therapeutic materials	[52]
β	hydroxytyrosol	^13^C, 2D ^1^H -^13^C HETCOR MAS	structure characterization, comparison of complexation methods	SEM	2018	drug carrier, obtaining better stability and extended shelf-life	[53]
β	ibuprofen	^13^C, ^1^H, variable contact time cross-polarization VCT-CP MAS	molecular structure and dynamics	PXRD, 1H NMR in solution	2017	nanosponges	[54]
γ	efavirenz	^13^C CP MAS	structure characterization	TGA, PXRD, 1H NMR in solution	2021	potential drug carrier enhancing drug’s low aqueous solubility	[55]
β, hydroxypropyl- β	furazolidone	^13^C, 2D WISE ^1^H -^13^C MAS	-	TGA-DSC, SEM, PXRD, Raman spectroscopy	2020	potential drug carrier enhancing drug’s low aqueous solubility and stability	[56]
α, β	limaprost	^13^C MAS	structure characterization	Raman spectroscopy, PXRD	2016	drug carrier, obtaining better stability and extended shelf-life	[57]
β	limaprost	^2^H MAS	structure characterization, complex stability	-	2014	drug carrier, obtaining better stability and extended shelf-life	[58]
α	lipoic acid	^13^C MAS	structure characterization, explanation of the complex formation	PXRD, 1H NMR in solution, SEM, PXRD, FT-IR, Raman spectroscopy	2015	ligand stabilization through complexation with CD	[18]
β	lithium cation	^13^C CP MAS, ^7^Li MAS	determine the dynamics of ion transport	PXRD, EIS	2020	tunnel-like polymer electrolytes, to facilitate lithium–ion transport	[59]
α	lithium cation	^13^C, ^7^Li, ^13^C CP MAS	reveal unique structural features	PXRD	2018	polymer	[60]
α	lithium cation	^13^C, ^1^H, ^7^N, ^1^H -^13^C, ^2^H, ^7^Li-^7^Li 2D MAS	detailed study of Li+ dynamics in nanochannel	WXRD, PXRD	2014	polymer	[61]
α, β	menthol	^13^C CP MAS	structure characterization	vibrational circular dichroism	2020	flavor encapsulation in preserved food and cosmetics	[62]
γ	methotrexate	^13^C CP MAS/TOSS	structure characterization	PXRD, FT-IR, SEM	2020	potential drug carrier enhancing drug’s low aqueous solubility and stability, metal organic framework	[63]
hydroxypropyl-β	-	^13^C CP MAS	structure characterization, amorphous/crystalline verification	SEM, AFM, PXRD, 2C ROESY	2018	drug-carrying nanofibers, different excipients	[64]
α, γ: native and porous	-	^13^C MAS	structure characterization, native vs. porous structure verification	FT-IR, 1H NMR in solution, SEM, PXRD	2019	absorption and separation CD-tunnels	[65]
β	-	^13^C MAS	structure characterization	PXRD, NMR in solution	2013	drug delivery systems, nanospheres	[66]
β	-	^13^C VCT ^13^C CP MAS	molecular structure and dynamics	FT-IR-ATR, Raman spectroscopy	2012	nanosponges	[67]
β	naphthalene	^13^C MAS	complex formation	FT-IR, TGA	2018	grafted CD on activated carbon for ligand absorption	[68]
β	p-nitrophenol	^13^C MAS	confirmation of CD immobilization on silica surface	FT-IR, TGA, SEM, XRS, XRD, ROESY NMR in solution	2015	CD-grafted silica gel for ligand absorption	[69]
β	p-nitrophenol	^13^C MAS	confirmation of CD immobilization on silica surface	FT-IR, TGA, SEM, XRS, XRD	2015	CD grafted on hybrid silica for ligand absorption	[70]
β	norfloxacin	^13^C MAS	conformation of obtaining a new solid-state form of a ligand upon crystallization	PXRD, FT-IR, SEM	2013	potential drug carrier	[71]
α	-	^13^C MAS	differentiation and description of amorphous and crystalline form	SEM, XRD, FT-IR, DSC, PSD, TGA	2015	organic-compound-absorbent	[72]
	AT1R antagonists	^13^C CP MAS	structure characterization	2D NOESY	2020	potential drug carrier enhancing drug’s low aqueous solubility	[73]
β	ornidazole	^13^C MAS	structure characterization	SEM, FT-IR, PXRD, DSC, TGA, 1H NMR in solution	2020	polymer microspheres	[74]
β	perfluorobutyric acid	^13^C DP, CP, high-power ^1^H/^19^F decoupling MAS	confirmation of complex formation, structure characterization	PXRD, DSC	2014	pollutant-removing agent	[75]
β	perfluorooctonic acid	^19^F direct polarization (DP) and ^13^C cross polarization MAS	confirmation of complex formation, structure characterization	FT-IR, DSC-TGA, PXRD	2013	pollutant-removing agent	[76]
2-hydroxypropylo- β	caffeinic and rosmarinic acid	^13^C CP MAS	structure characterization	ITC, mass spectrometry, 1H NMR in solution	2021	optimize pharmaceutic profile, enhance stability of natural food additives	[77]
β	CD modification with phosphorus groups	^31^P, CP MAS	structure verification with regard to other applied methods	ICP-OES, TGA, PCFC	2020	CD modification, promising matrices for environmentally benign fire-resistant coatings (lower combustibility)	[78]
β	pindolol	^13^C CP MAS	confirmation of CD-including polymer formation	TGA, 1H NMR in solution, MALDI-TOF MS	2020	polymer–drug conjugates	[79]
α, methylo-α, β, γ, γ50	poly(lactic acid)	^13^C CP MAS	confirmation of complex formation, possible complex ratio	1H NMR in solution, DSC	2018	polymer with poly(l-lactic acid) used as a packaging material, also for medical equipment	[80]
β, methoxy-azido-β, heptakis-(6-deoxy)-(6-azido)-β	calixarene	^13^C CP MAS, LGFS	confirmation of complex formation	FT-IR, TGA, porosimetry	2016	obtain pH-tunable nanosponges, mixed cyclodextrin-calixarene co-polymers	[81]
β	water organic pollutants	^13^C MAS	structure characterization	water contact angle, SEM, FT-IR, TGA, nitrogen adsorption–desorption isotherms, elemental analysis	2019	porous *CD*/pillar [5]arene copolymer for rapid removal of organic pollutants from water	[82]
β	p-nitrophenol	^13^C CP MAS	structure characterization	FT-IR, TGA, DSC, SEM, elemental (C and H) microanalyses	2011	microsphere polymeric materials with poly(acrylic) acid, improve sorption characteristics in aqueous environment	[83]
β	polyaniline	^13^C CP MAS	elucidate the inclusion effect on the dynamic structure of polyaniline	UV-Vis	2012	isolate polymer chains in a bulk system	[84]
β	methylene blue dye	^13^C MAS	confirmation of complex structure	SEM, FT-IR, TGA, water-contact-angle measurement	2018	removing agent, CD-based polymer containing carboxylic acid groups	[85]
2-hydroxy-propylo- β, α	carvedilol	^13^C MAS	structure characterization	XRPD, FT-IR, DSC	2020	CD-based poly(pseudo)rotaxanes, supramolecular gel	[86]
β	-	^13^C MAS	comparison of CD hydrate and CD polymer	UV-Vis, FT-IR, TGA, 1H NMR in solution	2016	CD-based polyurethanes	[87]
β, methyl-β, hydroxypropyl-β	praziquantel	^13^C MAS	structure characterization	PXRD	2015	potential drug carrier enhancing drug’s low aqueous solubility	[88]
2-hydroxypropylo- β	quercetin	^13^C MAS	structure characterization	1H NMR in solution	2016	potential drug carrier enhancing drug’s low aqueous solubility	[89]
hydroxypropyl-β	ripovacaine	^13^C CP MAS	prove the presence of the CD coating and ligand incorporation into CD	SEM, 1H NMR in solution	2014	drug carrier for local and prolonged delivery, crosslinked polymer with CD	[90]
β	sertraline	^13^C CP MAS	structure characterization and estimation of complex ratio	FT-IR, PXRD, Raman spectroscopy	2015	potential drug carrier enhancing drug’s low aqueous solubility	[91]
2-hydroxypropylo-β	silibinin	^13^C CP, ^1^H MAS	structure characterization	DSC, 2D NOESY and DOSY NMR, mass spectroscopy	2015	potential drug carrier enhancing drug’s low aqueous solubility and protecting against conjugation and metabolic inactivation	[92]
α, β, γ	cyclic siloxanes	^13^C, ^29^Si MAS	structure characterization	NMR in solution, FT-IR, TGA, PXRD, SEM–EDS, elemental analyses	2012	selective impurities removing agent	[93]
2-hydroxypropylo-β, β	sumatriptan	^13^C, ^1^H MAS	structure characterization	FT-IR, DSC	2018	potential drug carrier enhancing drug’s low aqueous solubility	[94]
β, permethylated- β	(2,20-dipyridylamine)chlorido(1,4,7 trithiacyclononane)ruthenium(II) chloride	^13^C CP MAS	structure characterization	PXRD, TGA, FT-IR, elemental analysis	2017	potential drug carrier enhancing drug’s low aqueous solubility	[95]
β	thiurams	^13^C CP MAS	structure characterization, encapsulation evidence	PXRD, TGA	2022	aqueous solubility increase	[96]
Mono-2-tosylated α, β, γ	N-tosylimidazole	^13^C CP MAS	structure characterization	XRD, mass spectrometry, diffuse reflectance UV−vis	2015	modified CD: mechanochemistry research	[97]
γ	valsartan	^13^C CP MAS	structure characterization	SR-FT-IR, PXRD, DSC, SAXS	2019	potential drug carrier enhancing drug’s low aqueous solubility, cyclodextrin metal organic framework	[98]
β, randomly methylated β	zaleplon	^13^C CP MAS	structure characterization	DSC, PXRD, SEM	2012	potential drug carrier enhancing drug’s low aqueous solubility, with hypromellose and polyvinylpyrrolidone	[99]
β	ciprofloxacin	^13^C MAS	structure characterization	cryo-SEM, FT-IR	2016	supramolecular CD hydrogel	[100]
α, β	cyclosporine A	^13^C CP MAS	structure characterization	PXRD, circular dichroism spectroscopy	2018	microspheres based on CD-polymers, potential drug carrier enhancing drug’s low aqueous solubility and bioavailability	[101]

**Figure 3 ijms-24-03648-f003:**
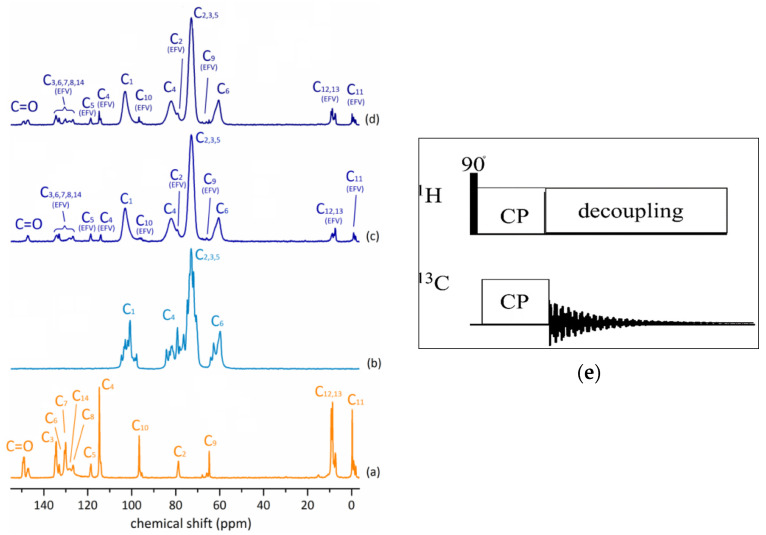
^1^H → ^13^C CP-MAS NMR spectra of (**a**) Efavirenz (EFV), (**b**) γ-CD, (**c**) (γ-CD)_3_@(EFV)_2_, and (**d**) γ-CD@EFV. Adapted from [55], licensed under CC BY 4.0. (**e**) Image on the right presents ^1^H → ^13^C CP NMR pulse sequence.

### 3.1. Direct Polarization versus Cross Polarization Experiments

CP may seem to be an obvious choice to study the CD-based materials, as it takes advantage of the properties of the protons coupled to the carbons, namely faster relaxation, which allows for reduction of the recycle delay and increases the signal-to-noise intensity in the spectrum at a constrained experimental time. The double irradiation process is used to transfer some of the proton’s magnetization to the carbon atoms. However, in complexes with a ligand of low protonation level, such as sodium perfluorooctane [41], perfluorobutyric acid [75], and perfluorooctanoic acid [76], direct polarization (DP) was applied. DP experiments were found to be useful in those cases, as they allowed for an increase in the ratio of guest-to-host signal intensities. In addition, as ^13^C CP/MAS NMR relies on the transfer of magnetization from protons or other abundant nuclei to ^13^C via the dipolar coupling mechanism, whereas the direct single pulse method does not, the comparison of DP and CP experiments may be very beneficial in some cases, as different information from ^13^C CP/MAS and single pulse ^13^C MAS spectra can be obtained.

Another type of measurement is the variable contact time (VCT) CP experiment; however, it is rarely applied. In the CD-based materials and complexes, the particular systems may differ in the proton spin–lattice relaxation time in the rotating frame (T_1ρ_) values. Therefore, by recording the ^1^H → ^13^C CP/MAS NMR spectra with various contact times, it is possible to obtain the collection of spectra in which the relative intensities of signals originating from different phases will vary. This allows one not only to detect the number of phases but also to selectively enhance or suppress the chosen signals, which can help in the signal assignment. VCT experiments found their particular application in the studies of nanosponges without [67] and with a guest [54] (more on nanosponges, please find in Section 4.2 Nanosponges and functionalized CDs). In those cases, not only the structure but also the dynamics of the systems have been analyzed, the latter enabled by the VTC NMR. To quote the already cited ibuprofen study [54]: ‘the outcome of VCT data processing is a “dynamic fingerprint” that can be used for the characterization of the polymer-drug system’. The CP-VCT approach is especially useful to compare loaded and not-loaded polymer systems when a structural difference between those two is not significant. This has been used to analyze the CDs which were parts of a polymer in the two studies cited above: CD vs. CD in nanosponges [67] or CD in nanosponges vs. CD in nanosponges with a guest [54].

### 3.2. Nuclei Other Than ^13^C

Due to the elemental composition of CDs, the 13C ssNMR experiments are the most common. Far less popular approaches are ^1^H NMR [20,28,41,43,61,92,94] and ^2^H NMR. The latter is used to analyze in detail a chosen part of a system. The only two examples for the CD-including systems are: CD complex with limaprost [57,58] and CD-including nanochannels for lithium ions [61]. The latter example shows the 2H NMR applicability in systems of high complexity, in this case, composed of polyethylene oxide and α-CD-forming tunnels. It should be noted, however, that solid-state spectra for other nuclei, such as ^7^Li, ^15^N, or ^31^P have been also recorded. The choice of the studied nuclei depends on the composition of the guest molecule in the complex. In some cases, the spectra of nuclei other than ^13^C or ^1^H can be particularly helpful, as they allow one to determine the guest:host ratio in the solid state or detect the presence of various solid phases. This is possible, as the chemical shifts of the nuclei originating from the complexed and “free” guest molecules usually differ.

### 3.3. Relaxation Studies

A relatively often-performed analysis is linking the spin–lattice relaxation times (T_1_) with the dynamical mobility. For instance, in the already mentioned CD-limaprost study [57,58], where 2H NMR has been used, the T_1_ measurements of deuterium atom allowed the conclusion that CD-constrained water molecules mobility is a main factor for the solid-state CD-limaprost complex stabilization. Another example is a temperature-dependent analysis of the CD-polyaniline complex [84], where ssNMR was used to elucidate the inclusion’s effect on the structure of polyaniline polymer. T_1_ values revealed that the guest’s inclusion into the CD’s cavity induces acceleration of the twisting motion of polyaniline chain, that the CD inclusion weakens intermolecular π−π interaction, and that this enhances the accompanying twisting motion.

### 3.4. Two-Dimensional Experiments

Structural information on CD-inclusion complexes can be obtained using 1D solid-state NMR experiments that have been applied to studies of crystalline and amorphous CD-based materials. However, in the reviewed literature, singular cases of 2D WISE (two-dimensional WIde-line SEparation), 2D HETCOR (two-dimensional HETeronuclear chemical shift CORrelation experiments), GCOSY (Gradient Correlation SpectroscopY), and GHSQC (Gradient Heteronuclear Single Quantum Coherence Spectroscopy) application are present, the latter two in one article [102]. WISE has been used in cases when an especially high risk of unseparated signals in a spectrum was predicted [56]. The 2D WISE NMR approach was applied for analysis of site-specific dynamics of the CD-including polymers: polyrotaxanes [19]. According to the authors of the study, it proved a dynamic difference between CDs and polyethylene glycol chains as well as the influence of chemical modifications on the polymer side chains.

## 4. ssNMR Applicability for Analysis of the Cyclodextrin Complexes

In the previous section, various ssNMR approaches used towards CD-including complexes published within last 10 years have been described. The rest of this review refers to the results of ssNMR application in such systems. This description has been divided into sections on the basis of the reasons for CD’s usage. General information has been gathered in Table 1.

### 4.1. Drug Carriers

The application of CDs as drug carriers is the most popular one (Figure 3). In most of the cases, ssNMR is used to prove the complexation, characterize the structure of a formed complex, and find a possible reason for its stability. However, as the usage of CDs serves to enhance the solubility, hence, bioavailability, of medical substances in solid-state forms of medication, a common reason for ssNMR experiments is either differentiation between amorphous and crystalline complex forms [22,23] or comparison of different complexation methods. The examples are the following complexes: CD-bisacodyl, where co-crystallization and co-evaporation have been compared [29], or CD-hydroxytyrosol, where comparison for physical mixing, spray-draying, and freeze-drying methods has been conducted [53]. Another reason to use ssNMR for systems where CD plays a role of a drug carrier is to determine, or at least estimate, the CD-guest ratio, as in case of the CD-sertraline [91] and CD-praziquantel [88] complexes.

An interesting example is the publication regarding the CD-furazolidone complex [56], where all the above-mentioned reasons can be found: thanks to ssNMR, complex formation has been proven, lyophilization has been found more effective than knitting, CD-guest molar ratio has been suggested, and the observed signal-broadening effect pointed out the loss of crystallinity upon complexation.

Another interesting example of the application of ssNMR was the study of the albendazole:succinyl-β-CD inclusion complexes in spray-dried samples [22]. Albendazole itself was found to exist in a few tautomeric forms, depending on its crystal structure. By comparing the chemical shifts present in the spectrum of a complex with the spectra of various forms of albendazole, it was possible to determine the ratio of particular tautomers in the studied complex.

#### Drug Carriers with Cellulose

A specific group of the CD-including drug carriers form systems with cellulose: CD-grafted cellulose fibers or hydrogels. Their purpose is either to be used as antimicrobial products with controlled drug release or to enhance solubility more than just by complexation with a CD. So far, all the published articles that address CDs grafted on cellulose refer to antimicrobic guests: ciprofloxacin hydrochloride [33], acetylsalicylic acid [36], ketoconazole [37,38,39], ciprofloxacin, doxorubicin, and paclitaxel [25]. Here, except for the mere characterization of the complex’s structure, ssNMR’s primary use is to prove that a process of CDs’ grafting onto cellulose has been successfully completed.

### 4.2. Nanosponges and Functionalized CDs

Nanosponges are polymers, porous nanoparticles used as toxin-removing agents or drug delivery systems [101]. Historically, nanosponges were referred to as ‘cyclodextrin nanosponges’ because those with CDs were the first ever obtained. Nanosponges are composed of CDs and cross-linking agents. With regard to nanosponges, ssNMR is used to confirm the system’s formation, analyze the system’s structure, prove the guest’s incorporation into CD’s cavity, and, as it has been mentioned in Section 2 (ssNMR approaches used to), to differentiate between loaded and not-loaded systems. However, there are more examples. For instance, in [64], the effect of different excipients is measured by the relaxation time of carbons according to the principle that the more mobile the carbon atom, the slower it relaxes; this is, in turn, a plasticizing effect of an excipient. The historical ‘cyclodextrin nanosponges’ were composed of native CDs. Today, experiments with substituted CDs are performed. For example, in a pH-tunable nanosponge system composed of a cyclodextrin–calixarene mixture, three different CDs (β-CD, methoxy-azido-β-CD, and heptakis-(6-deoxy)-(6-azido)-β-CD) have been tested [81]. ssNMR analysis showed that methylation of CD favors its incorporation in the polymeric material, likely due to an increase in its hydrophobic character.

Not only nanosponges are CD-polymers-based systems; there are also microspheres. As indicated in [103], ssNMR can be used to define the inner structure of such a system. In this particular study, a physical mixture of CD-polymer and cyclosporin A was compared with spherical amorphous solid dispersion of the co-polymer.

Another aspect is research of CD’s functionalization, in other words, analysis of changes applied solely to CDs. As an example, it can serve β-CD modification with phosphorus groups. This allows such CDs to serve as promising matrices for environmentally benign fire-resistant coatings [78]. Due to phosphorus presence, for structure verification, 31P ssNMR has been applied. Another interesting case is tosylation of CDs [97]. In this experiment, N-tosylimidazole forms covalent bonds with the CD, and, at the same time, N-tosylimidazole’s aromatic ring is incorporated into the CD’s cavity. In order to probe spatial proximity between the guest and CD, 1H−1H double-quantum/single-quantum (DQ-SQ) spectroscopy was employed.

### 4.3. Removing Agents and Sensory Devices

While its role as a drug carrier is the CD complex’s most common application, the second most common is its role as a removing agent. Such water pollutants are toxins, such as p-nitrophenol [69,70], naphthalene [68], or endocrine-disrupting chemicals, including bisphenol A and other estrogens [30,35,47]. Such systems may be formed out of silica gel CD-including matrices as well [69]. A wide range of ssNMR usage results can be obtained for those systems: from structure characterization and confirmation of the guest incorporation through the differentiation between system’s crystallization methods up to ‘crystalline or amorphous’ statement. In other words, ssNMR enables a complex analysis of a complex itself and delivers information about the complex formation. For example, in one of the p-nitrophenol-removing studies [83], it was shown that the reduced intensity of one of the carbonyls occurred due to esterification of the carbonylic acid of a polyacrylic acid while making a bond with a CD. Moreover, in contrast to sharp ssNMR signals of hydrated CD, broad resonance lines on the spectrum after system’s preparation indicate the amorphous character of the generated material.

A thematically separated, but technically similar topic, is CDs’ application as sensory devices for ions [20,21,28]. There, ssNMR serves to prove whether a selected ion incorporation took place.

### 4.4. Other Application

CD complexes in solid state are used also as enantioseparators [32], food and flavor-encapsulating agents [62,96], as part of the antimicrobial packaging materials [19,46], etc. Thanks to ssNMR, the inner structure, complexation process, CD-guest molar ratio, etc., are defined.

Interesting example are nanoporous materials consisting of CD tunnels, often in the form of polyrotaxanes, where a rotaxane is a mechanically (i.e., without covalent bonds) interlocked molecule consisting of strings and rings (in this case, rings are CDs) [104]. ssNMR experiments of a nanoporous material with different CDs [65] indicate that the material with γ-CD forms a more rigid structure in contrast to other analyzed CDs. There, the splitting of the carbon atom is restricted due to formation of a tunnel-type crystalline system. According to the article published in 2019 [65], this is characteristic of highly cross-linked CDs and polyrotaxanes and can be observed by application of ssNMR.

### 4.5. The Limitations and the Difficulties of ssNMR Approaches in the Studies of CD-Based Materials

Despite all of the advantages and unique opportunities offered by the ssNMR analysis, to maintain the research integrity, the limitations and the difficulties of reviewed approaches in the studies of CD-based materials also must be emphasized. ssNMR is not a perfect tool to study the CD-including systems, as such a method does not exist and, like any other analytical technique, has its disadvantages.

First, the anticipated changes in the ssNMR spectra of the studied materials may not be detectable, or the differences may be small and ambiguous. This can be caused by several reasons, starting from the weakness of interactions between the host and guest molecule, resulting in only slight differences in the chemical shifts between the corresponding atoms of complexed and non-complexed guest molecules. Another reason is associated with the typical composition of the sample. In the ^13^C ssNMR spectra of cyclodextrins, both in free and complexed forms, many signals in the wide rage 50–100 ppm can be observed. It is, thus, very likely that the signals originating from the guest molecules may be overlapped with those from the cyclodextrin. This, in some cases, may be solved by setting particular values of either CP contact time or recycle delay to selectively increase the intensity of the signals originating from the guest. However, in most cases, the differences between the relaxation of host and guest are not significant, which hampers such an approach. For example, while studying the loading of γCD-MOF by MTX, the authors have not observed the noticeable shifting of the cyclodextrin signals, explaining that this observation points out that the framework has no strong interactions with the drug molecules [63]. On the other hand, the ^13^C NMR data were consistent with the conclusions made by the authors based on the FTIR experiments.

Another aspect is associated with solid state of matter and is often encountered when comparing the ssNMR and NMR analysis in other phases, such as in solutions [105] or supercritical fluids [106]. In the NMR spectra of solution samples, fast molecular motions lead to the averaging of individual contributions to the hamiltonian, whereby only the isotropic value of the chemical shift and J coupling is observed. In the case of samples in solid phase, inhibition of molecular motions leads to restoration of the full hamiltonian of interaction, resulting in a significant increase of the width of signal in the spectrum, often exceeding its spectral range. In order to narrow the line of the NMR spectrum in the solid state, a magic-angle spinning (MAS) is used. It involves rapid rotation of the sample at a specific angle to the external magnetic field. If the angle is approximately 54o, the lines become narrowed due to the physical averaging of the tensor of anisotropy of chemical shift to the isotropic value as well as the reduction of dipole interactions. The probe heads, usually used for measuring in the solid state, allow the rotation with a frequency of 15 kHz; yet, this is often insufficient to completely eliminate the dipolar couplings, particularly in the case of interaction with the hydrogen nuclei. Therefore, during the signal collection, further decoupling of these interactions with the use of high-powered pulses is used.

There are also some technical issues that hinder the application of ssNMR; in general, the relatively high cost of the analysis and long experimental time needed to obtain good quality results, especially in comparison with the FT-IR or PXRD.

## 5. Conclusions

Although ssNMR is not the most commonly used analytical method to characterize CDs and CD-based materials, it is likely one of the most versatile and comprehensive. It can be used to study the solid-state materials with all possible long-range order, amorphous, polymorphous, microcrystalline, polycrystalline, or crystalline. In addition, it can be readily applied to study both single-phase materials as well as mixtures and multi-component systems. The number of ssNMR experiments that can be performed for a particular sample is enormous—starting from the most common 13C CP MAS, but, depending on the type of host molecule, analysis of the other nuclei is also possible. Further, due to the non-covalent interactions occurring between the host and guest molecules in the CDs complexes, each of the components usually preserves their particular proton spin–lattice relaxation time in the rotating frame (T1ρ), which provides an opportunity to benefit from the variable contact time CP experiments. Additionally, the variable temperature experiments and relaxation measurements can provide the information on both structure and dynamics of the studied system. Moreover, in some cases, the reaction kinetics, i.e., the activation energy, can be determined on the basis of the ssNMR analysis.

From the practical point of view, ssNMR can be used to confirm the inclusion process, determine the guest:host ratio, assess the number of phases in the solid sample, detect the non-complexed “free” guest and host molecules, and compare the degree of crystallinity. As in the NMR spectra, each signal can be assigned to a particular atom, and it is possible to determine which part of the guest molecule is involved in the formation of an intermolecular force with a CD molecule. In addition, comparison of the chemical shift value originating from the complexed and “free” guest molecule allows it to provide the information on the character of this interaction. Moreover, ssNMR analysis is likely the most convenient way to study how the method of complex preparation affects the structure of the final product.

Due to the limited number of analytical methods that can be applied to study the CD-based materials in the solid state, it is not surprising that the number of works describing successful applications of ssNMR in this field has been increasing constantly for the last 30 years. Assuming that the accessibility of ssNMR analysis is increasing, together with the constant development of this method, it is very likely that soon ssNMR will become a standard approach in the analysis of all types of solid CD-based materials.

## Figures and Tables

**Figure 1 ijms-24-03648-f001:**
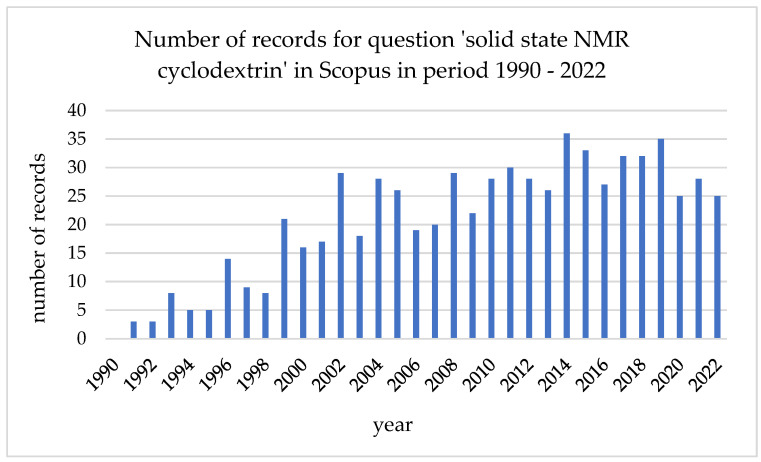
Number of records for query ‘solid-state NMR cyclodextrin’ in Scopus in period 1990–2022.

**Figure 2 ijms-24-03648-f002:**
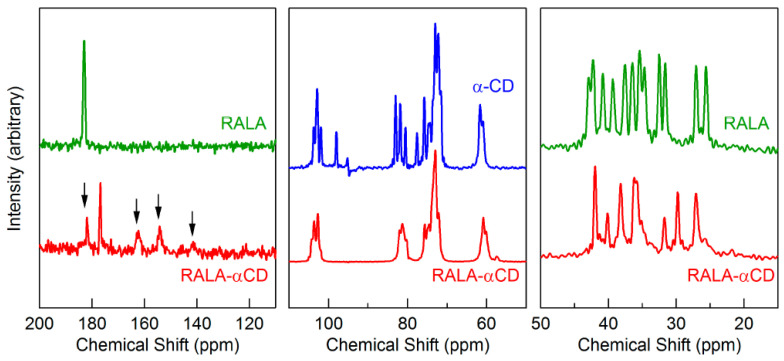
^13^C-CP/MAS NMR spectra of the crystalline R(+)-α-Lipoic Acid (RALA)-αCD complex, free α-CD, and free RALA, recorded at 6 kHz spinning frequency. The arrows indicate the spinning side bands Adapted from [18], licensed under CC BY 4.0. [18].

## Data Availability

Not applicable.

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
