# Peer review of "A Review of Applications of Solid-State Nuclear Magnetic Resonance (ssNMR) for the Analysis of Cyclodextrin-Including Systems"

_ijms, 2023, doi:10.3390/ijms24043648_

Round 1

Reviewer 1 Report

 In this review the examples of  characterize cyclodextrins (CDs) and CDs including systems by  solid state nuclear magnetic resonance
 (ssNMR) spectroscopy. ssNMR is a very sensitive method to study the chemical structure of the CDs by monitoring the  changes in the resonance frequencies of protons in the CD cavity. The NMR spectrum of CD can be used as a fingerprint to identify features of the structure of CD.  But at the same time with using ssNMR spectroscopy it is not easy to determine the structure and the number of molecules in a CD complex. For example as you can see in ref. 71. Kritskiy, I.; Volkova, T.; Sapozhnikova, T.; Mazur, A.; Tolstoy, P.; Terekhova, I. Methotrexate-loaded metal-organic frame- 567
works on the basis of γ-cyclodextrin: Design, characterization, in vitro and in vivo investigation. Materials. Sc. And Eng. 568
C 2020, 111, 110774, doi.org/10.1016/j.msec.2020.110774
the loading of γCD-MOF by MTX does not result in the noticeable shifting of the cyclodextrin signals. You can see the same situation for other CD systems. It follows from this that the solid-state NMR spectroscopy is a very specific tool for the characteristic of CD.
Often CD complexes are obtained in solution, and it is logical to use NMR spectroscopy in solution. The authors should compare the NMR methods in solutions [Garibyan A., Delyagina E., Agafonov M., Khodov I., Terekhova I.Effect of pH, temperature and native cyclodextrins on aqueous solubility of baricitinib (2022) Journal of Molecular Liquids, 360, art. no. 119548, DOI: 10.1016/j.molliq.2022.119548] and supercritical fluids [Ivanova, G.I., Vão, E.R., Temtem, M., Aguiar-Ricardo, A., Casimiro, T., Cabrita, E.J. High-pressure NMR characterization of triacetyl-β-cyclodextrin in supercritical carbon dioxide (2009) Magnetic Resonance in Chemistry, 47 (2), pp. 133-141.  DOI: 10.1002/mrc.2365
] with experiments in a solid phase. It is necessary to describe the limitations of ssNMR approaches and the difficulties that you may encounter when using these methods.
Based on the issues above, I have to recommend revising the manuscript after significant changes to  the current version of this manuscript. Hope the authors could make an improvement and re-submit the revised version later.

Author Response

Comment:

In this review the examples of  characterize cyclodextrins (CDs) and CDs including systems by  solid state nuclear magnetic resonance  (ssNMR) spectroscopy. ssNMR is a very sensitive method to study the chemical structure of the CDs by monitoring the  changes in the resonance frequencies of protons in the CD cavity. The NMR spectrum of CD can be used as a fingerprint to identify features of the structure of CD.  But at the same time with using ssNMR spectroscopy it is not easy to determine the structure and the number of molecules in a CD complex. For example as you can see in ref. 71. Kritskiy, I.; Volkova, T.; Sapozhnikova, T.; Mazur, A.; Tolstoy, P.; Terekhova, I. Methotrexate-loaded metal-organic frame-  works on the basis of γ-cyclodextrin: Design, characterization, in vitro and in vivo investigation. Materials. Sc. And Eng. 568 C 2020, 111, 110774, doi.org/10.1016/j.msec.2020.110774 the loading of γCD-MOF by MTX does not result in the noticeable shifting of the cyclodextrin signals. You can see the same situation for other CD systems. It follows from this that the solid-state NMR spectroscopy is a very specific tool for the characteristic of CD.

Often CD complexes are obtained in solution, and it is logical to use NMR spectroscopy in solution. The authors should compare the NMR methods in solutions [Garibyan A., Delyagina E., Agafonov M., Khodov I., Terekhova I.Effect of pH, temperature and native cyclodextrins on aqueous solubility of baricitinib (2022) Journal of Molecular Liquids, 360, art. no. 119548, DOI: 10.1016/j.molliq.2022.119548] and supercritical fluids [Ivanova, G.I., Vão, E.R., Temtem, M., Aguiar-Ricardo, A., Casimiro, T., Cabrita, E.J. High-pressure NMR characterization of triacetyl-β-cyclodextrin in supercritical carbon dioxide (2009) Magnetic Resonance in Chemistry, 47 (2), pp. 133-141.  DOI: 10.1002/mrc.2365] with experiments in a solid phase. It is necessary to describe the limitations of ssNMR approaches and the difficulties that you may encounter when using these methods.

Based on the issues above, I have to recommend revising the manuscript after significant changes to  the current version of this manuscript. Hope the authors could make an improvement and re-submit the revised version later.

Response:

Thank you very much for the significant effort needed to create this review. We’ve found all of your suggestions very helpful in improving the quality of our manuscript. We agree that the ssNMR is not a perfect tool to study the CDs including systems, as such a method does not exist. Like any other analytical tools, ssNMR has its advantages and disadvantages. After deeply discussing your comments, we have decided to introduce another paragraph, at the end of the Discussion part, presenting  the limitations of ssNMR approaches and the difficulties that you may encounter when using these methods, in particular comparison with the NMR methods in solutions and supercritical fluids.

4.5. The limitations and the difficulties of ssNMR approaches in the studies of CDs-based materials.

Despite all of the advantages and unique opportunities offered by the ssNMR analysis, to maintain the research integrity, also the limitations and the difficulties of reviewed approaches in the studies of CDs-based materials must be emphasized. ssNMR is not a perfect tool to study the CDs including systems, as such a method does not exist and, like any other analytical technique, has its disadvantages.

First, the anticipated changes in the ssNMR spectra of the studied materials may not be detectable or the differences may be small and ambiguous. This can be caused by several reasons, starting from the weakness of interactions between the host  and guest molecule resulting in only slight differences in the chemical shifts between the corresponding atoms of complexed and non-complexed guest molecule. Another reason is associated with the typical composition of the sample. In the 13C ssNMR spectra of cyclodextrins, both in free and complexed forms, a lot of signals in the wide rage 50-100 ppm can be observed. It is very likely than, that the signals originating from the guest molecules may be overlapped by those from the cyclodextrin. This, in some cases, may be solved by setting particular values of either CP contact time or recycle delay, to selectively increase the intensity of the signals originating from the guest. However, in most cases the differences between the relaxation of host and guest are not significant, which hampers such approach. For example, while studying the loading of γCD-MOF by MTX the authors have not observed the noticeable shifting of the cyclodextrin signals. explaining that this observation points out that framework has no strong interactions with the drug molecules [Kritskiy, I.; Volkova, T.; Sapozhnikova, T.; Mazur, A.; Tolstoy, P.; Terekhova, I. Methotrexate-loaded metal-organic frame-  works on the basis of γ-cyclodextrin: Design, characterization, in vitro and in vivo investigation. Materials. Sc. And Eng. 568 C 2020, 111, 110774, doi.org/10.1016/j.msec.2020.110774]. On the other hand, the 13C NMR data were consistent with the conclusions made by the authors, based on the FTIR experiments.

Another aspect is associated with that solid state of matter and is often encountered when comparing the ssNMR and NMR analysis in other phases, such as in solutions [Garibyan A., Delyagina E., Agafonov M., Khodov I., Terekhova I. Effect of pH, temperature and native cyclodextrins on aqueous solubility of baricitinib (2022) Journal of Molecular Liquids, 360, art. no. 119548, DOI: 10.1016/j.molliq.2022.119548] or supercritical fluids [Ivanova, G.I., Vão, E.R., Temtem, M., Aguiar-Ricardo, A., Casimiro, T., Cabrita, E.J. High-pressure NMR characterization of triacetyl-β-cyclodextrin in supercritical carbon dioxide (2009) Magnetic Resonance in Chemistry, 47 (2), pp. 133-141.  DOI: 10.1002/mrc.2365]. In the NMR spectra of solution samples, fast molecular motions lead to the averaging of individual contributions to the hamiltonian, whereby only the isotropic value of the chemical shift and J coupling is observed. In the case of samples in solid phase, inhibition of molecular motions leads to restoration of the full hamiltonian of interaction, resulting in a significant increase of the width of signal in the spectrum, often exceeding its spectral range. In order to narrow down the line of the NMR spectrum in the solid state, a magic-angle spinning (MAS) is used. It involves rapid rotation of the sample at specific angle to the external magnetic field. If the angle is about 54o the lines get narrowed down due to the physical averaging of the tensor of anisotropy of chemical shift to the isotropic value as well as the reduction of dipole interactions. The probe heads usually used for measuring in the solid state allow the rotation with a frequency of 15 kHz; yet, this is often insufficient to completely eliminate the dipolar couplings, particularly in the case of interaction with the hydrogen nuclei. Therefore, during the signal collection, further decoupling of these interactions with the use of high-power pulses is used.

There are also some technical issues, that hinder the application of ssNMR in general, that is the relatively high cost of the analysis and long experimental time needed to obtain the good quality results, especially in comparison to the FT-IR or PXRD.

Author Response

Comment:

I have read with great interest the submitted manuscript “Application of solid state nuclear magnetic resonance (ssNMR) for the analysis of cyclodextrin-including systems.” by Mazurek and Szeleszczuk on the application of SNMR on the CD and CD systems. The work is of interest to many research groups working on this and similar host-guest systems. I do recommend publication of this manuscript with a minor modification which I think will enrich the work.

Response:

Thank you very much for the significant effort needed to create this review. We’ve found all of your suggestions very helpful in improving the quality of our manuscript. Below, please find the direct responses to your comments.

Comment:

Would be helpful for a reader that the title of the article be slightly modified; I suggest to add “A review of ………” at the start of the title: “A review of application of solid state nuclear ……..”

Response:

Thank you, that’s a great idea! The title has been changed to “A review of applications of solid state nuclear magnetic resonance (ssNMR) for the analysis of cyclodextrin-including systems.”.

Comment:

The authors provided a well written introduction. I would like to have a small paragraph between line 71 and 72; elaborate on the type of new results obtained by NMR in comparison with the other methods. Which physics parameters are well obtained by NMR that are not fully supported by other methods?

Response:

As requested by the Reviewer, a short paragraph between lines 71 and 72 has been added:

“In particular, ssNMR can provide the information on orientation of the guest molecule inside the cavity and the complex stability in the solid state. It also enables the quantitative analysis of the phases, especially the complexed and non-complexed guest molecules. Besides, this technique allows to study the local molecular dynamics of a guest molecules and the nature of intermolecular interactions between the host and the guest.”

Comment:

Line 93: please change doesn’t to “does not” and do the same elsewhere (line 97)

Response:

This has been corrected, as instructed by the Reviewer.

Comment:

Figure 2:  I assume the presented results is for proton-carbon CP using MAS. It is important to indicated the MAS speed within the caption..

Response:

The Reviewer is correct. The caption of this figure has been changed to: “|Figure 2. 113C CP/MAS NMR spectra of the crystalline R(+)-α-Lipoic Acid (RALA)-αCD complex, free α-CD, and free RALA, recorded at 6 kHz spinning frequency. The arrows indicate the spinning side bands [18].”.

Comment:

Section 3.1: although it is not much required, but including a CP NMR pulse sequence to this section would be a useful information.

Response:

The 113C CP NMR pulse sequence has been introduced in the form of a Figure 3.

Comment:

Table2: I assume the last column in this table that reads “No” refers to the “reference number”. Please write it as “Ref. #” instead of “No”.

Response:

Yes, the Reviewer is correct, this has been changed as instructed.

Comment:

Table 2: The same figure, please move all the contains within cells to the left (currently texts are at the center of each cell, move them to the left)

Response:

This has been changed, as instructed by the Reviewer.

Comment:

Section 4.4: consider changing the title to “Other applications” instead of “Others”

Response:

The title of Section 4.4. has been changed to “Other applications”, as instructed by the Reviewer.

Reviewer 3 Report

Very interesting review where the author has well described the use of solid state NMR on cyclodextin based inclusion complexes except he forgot to mention in paragraph 4.2 an article using this strategy ref.https://doi.org/10.3390/pharmaceutics10040285

Author Response

Comment:

Very interesting review where the author has well described the use of solid state NMR on cyclodextin based inclusion complexes except he forgot to mention in paragraph 4.2 an article using this strategy ref. https://doi.org/10.3390/pharmaceutics10040285

Response:

Thank you very much for a nice comment. In the revised version we have included the suggested work Lahiani-Skiba, M.; Hallouard, F.; Bounoure, F.; Milon, N.; Karrout, Y.; Skiba, M. Enhanced Dissolution and Oral Bioavailability of Cyclosporine A: Microspheres Based on αβ-Cyclodextrins Polymers. Pharmaceutics 2018, 10, 285. https://doi.org/10.3390/pharmaceutics10040285 both in the section 4.2. as well as in the Table 2.

Round 2

Reviewer 1 Report

The authors have provided an extensive literature review, which is well organized and provides an overview of the applications and implications of solid state NMR for cyclodextrin-including systems. The authors also provide details of the experiments and data analysis, which are clearly and concisely described. The authors also discuss the implications of the results and how they relate to the literature. Overall, the manuscript is of high quality and is suitable for publication in IJMS.